# Qualitative evaluation of barriers and facilitators to hepatocellular carcinoma care in North Carolina

**Emily M. Ray**[1,2]*, **Randall W. Teal**[2,3], **Jessica Carda-Auten**[2,3], **Erin Coffman**[4], **Hanna K. Sanoff**[1,2]

**1** Department of Medicine, Division of Oncology, University of North Carolina, Chapel Hill, North Carolina, United States of America, **2** Lineberger Comprehensive Cancer Center, University of North Carolina, Chapel Hill, North Carolina, United States of America, **3** Connected Health Applications and Interventions Core, University of North Carolina, Chapel Hill, North Carolina, United States of America, **4** Gillings School of Global Public Health, University of North Carolina, Chapel Hill, North Carolina, United States of America

* emily_ray@med.unc.edu

## Abstract

### Background

Many patients with hepatocellular carcinoma (HCC) never receive cancer-directed therapy. In order to tailor interventions to increase access to appropriate therapy, we sought to understand the barriers and facilitators to HCC care.

### Methods

Patients with recently diagnosed HCC were identified through the University of North Carolina (UNC) HCC clinic or local hospital cancer registrars (rapid case ascertainment, RCA). Two qualitative researchers conducted in-depth, semi-structured interviews. Interviews were audiotaped, transcribed, and coded.

### Results

Nineteen interviews were conducted (10 UNC, 9 RCA). Key facilitators of care were: physician knowledge; effective communication regarding test results, plan of care, and prognosis; social support; and financial support. Barriers included: lack of transportation; cost of care; provider lack of knowledge about HCC; delays in scheduling; or poor communication with the medical team. Participants suggested better coordination of appointments and having a primary contact within the healthcare team.

### Limitations

We primarily captured the perspectives of those HCC patients who, despite the challenges they describe, were ultimately able to receive HCC care.

**Data Availability Statement:** There are ethical restrictions on sharing a de-identified data set, as the data contain potentially sensitive information, including patient names, provider names, cancer

diagnosis, home residence city and state, and site of medical care, all of which could enable identification of a participant. The participant consent form indicates that we will share de-identified excerpts in research manuscripts but not the entire transcript. As such, we cannot share these data publicly. The contact for the research ethics committee at UNC is: ethicsandintegrity@unc.edu.

**Funding:** This research was partially supported by a National Research Service Award Post-Doctoral Traineeship from the Agency for Healthcare Research and Quality sponsored by The Cecil G. Sheps Center for Health Services Research, The University of North Carolina at Chapel Hill, Grant No. T32-HS000032 (EMR). https://www.ahrq.gov/funding/training-grants/rsrchtng.html This research was also supported by The Murphy Family Research Fund through the Lineberger Comprehensive Cancer Center (HKS). The funders had no role in study design, data collection and analysis, decision to publish, or preparation of the manuscript.

**Competing interests:** The authors have declared that no competing interests exist.

## Conclusions

This study identifies key facilitators and barriers to accessing care for HCC in North Carolina. Use of the RCA system to identify patients from a variety of settings, treated and untreated, enabled us to capture a broad range of perspectives. Reducing barriers through improving communication and care coordination, assisting with out-of-pocket costs, and engaging caregivers and other medical providers may improve access. This study should serve as the basis for tailored interventions aimed at improving access to appropriate, life-prolonging care for patients with HCC.

## Introduction

Hepatocellular carcinoma (HCC) is a major cause of morbidity and mortality in the United States and around the world, and the incidence is increasing [1–5]. Although life-prolonging and even curative therapies exist, depending on the stage of disease [4,6,7], work by our group (Sanoff HK, unpublished data) and others has shown that many HCC patients—upward of half among Veterans Affairs (VA) and Medicare populations—do not receive any cancer-directed therapy [8–10]. Even among patients with early stage HCC and without cirrhosis, treatment rates are surprisingly low [8–10].

While low treatment rates in HCC are well established, the patient- and system-level factors contributing to low treatment rates are not well described. Comorbidities, socioeconomic status, insurance status, social support, racial and ethnic disparities, and availability of multidisciplinary subspecialty consultation likely all play a role [8,9,11–19]. Thus, there are likely both patient and health system factors that determine access to and quality of HCC care.

In order to increase the number of patients receiving appropriate therapy for HCC, it is critical to understand the barriers and facilitators to accessing HCC care from the patient perspective. The primary aim of this study was to explore patients' perspectives on their initial diagnosis, their medical care team, navigating the healthcare system, and barriers to care.

## Patients and methods

### Participants and recruitment

We conducted 19 in-depth, individual interviews either in person or over the telephone with patients within 6 months of their initial HCC diagnosis between November 2017 and October 2018. We identified patients either through the University of North Carolina (UNC) multidisciplinary HCC clinic or the North Carolina Central Cancer Registry Rapid Case Ascertainment (RCA) system, a system in which local hospitals identify relevant cancer cases to allow early contact with patients for research purposes. Overall eligibility criteria were: ages 18 years or older, clinically or histologically confirmed diagnosis of HCC, within six months of the initial diagnosis of HCC, and willing and able to participate in an in-depth interview. Receipt of treatment for HCC was not required for participation. Within the HCC clinic, eligible patients were first identified through physician referrals, then approached by the study coordinator and informed about the study. We used a purposive non-probability sample in which the characteristics of individuals were used as the basis of selection, chosen to reflect the diversity and breadth of the sample population. In this case, physician referrals considered characteristics including age, race, marital status, location of home residence, and socioeconomic status to try and reflect the patients who come to UNC Health for HCC treatment and care [20]. Patients

identified through the RCA system were reported to the study coordinator on a monthly basis without consideration for patient characteristics. RCA patients and their doctors received a letter in the mail describing the study. Interested patients from the HCC clinic and RCA were contacted by the qualitative research team to further describe the study and obtain informed consent. Because potential subjects were located across the state of North Carolina and interviews could be performed in person or by telephone, in-person contact between the potential subject and study team was not required. As such, we obtained informed consent via telephone, using a script approved by our Institutional Review Board. Consent was documented on an IRB-approved form signed by the study coordinator. Additionally, at the time of the interview, patients were again read a script regarding the voluntary nature of participation and given the opportunity to decline participation before proceeding with the in-depth interview. Participants received a $40 gift card for participation. Recruitment continued until thematic saturation was reached, meaning no new insights or information are being found by the researcher and additional data collection would not likely yield new information [21].

This study was approved by the Institutional Review Board at the University of North Carolina at Chapel Hill.

### Data collection procedures

This exploratory qualitative study was guided by social determinants of health and access to healthcare conceptual frameworks to help identify facilitators and barriers to patient access to HCC care [22]. Prior to the interviews, basic demographic and clinical information was abstracted from the medical record or cancer registry. Investigators and the qualitative team collaborated in developing the semi-structured interview guide (S1 Appendix) to explore key areas including participant perceptions on their initial liver cancer diagnosis, their medical care team, provider communication, scheduling appointments, treatment plans, barriers to treatment, and daily functioning. These key areas were selected based on existing literature regarding barriers to care in HCC and the clinical experience of the investigators [8,10,11,16,19,23]. Interviews were conducted by experienced qualitative researchers (RT, JCA) and took place either in-person or by phone based on the preference of the participant. The interviews lasted approximately 60 minutes and were audio-recorded and transcribed by a professional service.

### Analysis

Transcripts were imported into Dedoose (a qualitative software management tool, version 7.6.21, SocioCultural Research Consultants, Los Angeles, CA). Directed content analysis was applied to code transcripts using a codebook developed from the interview guide and notes taken during data collection [24]. The initial codebook was independently applied by three senior qualitative researchers (RT, JCA, MV) to several transcripts. Codes were compared across the three researchers, and discrepancies were identified and reconciled through discussion and consensus. This heuristic process continued until replicability of coding occurred across researchers, at which point the codes and coding decisions were finalized. The final version of the codebook (S2 Appendix) was then applied to the remaining transcripts. After coding all transcripts, researchers generated a report for each code to create narrative summaries, identify emergent themes and sub-themes, and select illustrative quotes. All participant perspectives, even if just stated by one participant, were considered themes. Semi-quantification words like "most" and "some" provide readers with some indication of how often a concept or theme appeared across interviews but without sharing numbers that could be misinterpreted

as generalizable [25]. Findings were organized to identify patient-perceived facilitators and barriers to accessing and navigating HCC treatment and care.

## Results

Sixty-one eligible patients were identified. Ten patients from the UNC HCC clinic and 9 patients from the RCA system participated. Reasons for non-participation are shown in Fig 1. Demographic and clinical characteristics of the participants and non-participants are shown in Table 1 Notably, the majority of participants and non-participants had Child-Pugh A liver disease and stage A or B HCC. The key themes that emerged from the interviews are presented in four main areas: (1) HCC Diagnosis–Receiving the news; (2) Facilitators of HCC care; (3) Barriers to HCC care; and (4) Participant suggestions for improvement. The key themes and subthemes, along with representative quotes, are found in Table 2.

### HCC diagnosis–receiving the news

Most participants said their liver cancer diagnosis was made incidentally while seeking care for an unrelated health issue. Reactions to being diagnosed with liver cancer were mixed. While some participants were not surprised by the diagnosis (i.e., they stated they were "half-expecting" it based on known underlying liver disease), others were, in their words, "shocked." Some participants had strong emotional reactions to the timing and diagnosis of their liver cancer. In several instances, participants were confused and upset that their providers did not detect liver cancer earlier.

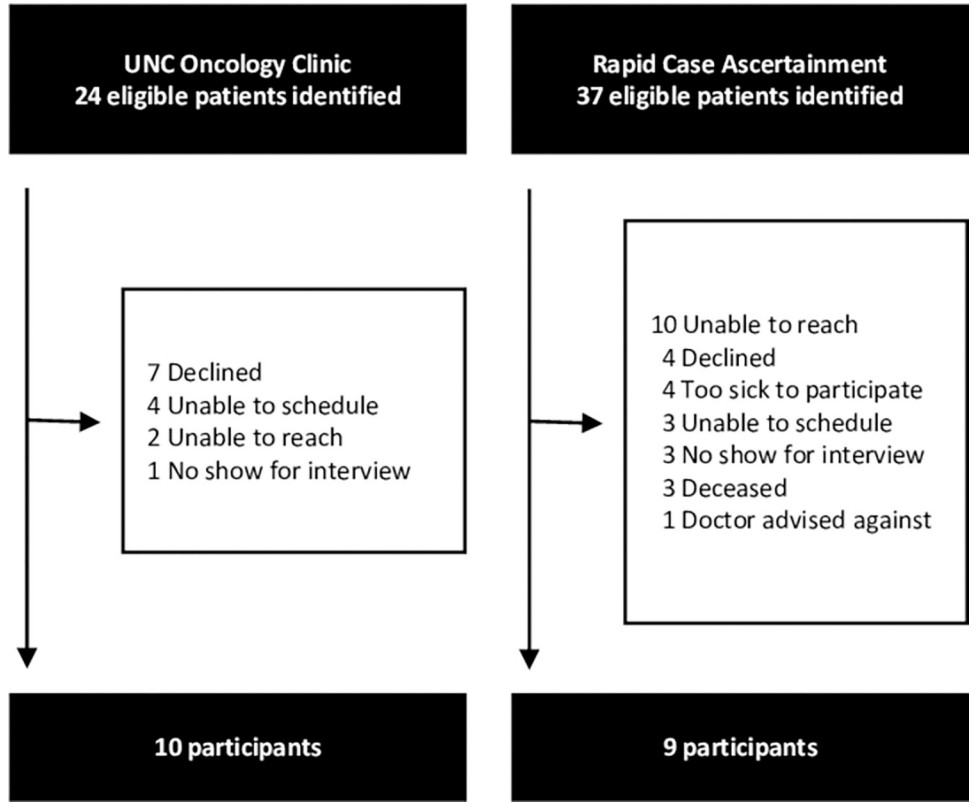

**Fig 1. Recruitment of HCC patients for qualitative interviews.** Patients were recruited from the UNC Multidisciplinary Oncology Clinic and the North Carolina Central Cancer Registry Rapid Case Ascertainment system.

**Table 1. Demographics and clinical characteristics of participants and non-participants.**

| | UNC Oncology Clinic | | Rapid Case Ascertainment | |
|---|---|---|---|---|
| | **Participants (n = 10)** | **Non-Participants (n = 14)** | **Participants (n = 8)** | **Non-Participants (n = 28)** |
| Age, in years, median (range) | 64 (51–78) | 64 (37–79) | 63 (39–82) | 68 (29–86) |
| **Race** | | | | |
| White | 9 (90%) | 10 (71%) | 5 (63%) | 20 (71%) |
| Black | 1 (10%) | 4 (29%) | 3 (37%) | 5 (18%) |
| Asian | 0 | 0 | 0 | 1 (4%) |
| Other / Unknown | 0 | 0 | 0 | 2 (7%) |
| **Sex** | | | | |
| Male | 7 (70%) | 10 (71%) | 7 (88%) | 20 (71%) |
| Female | 3 (30%) | 4 (29%) | 1 (12%) | 8 (29%) |
| **Stage (BLCL)[a]** | | | Staging information not available | |
| A | 3 (30%) | 6 (43%) | | |
| B | 6 (60%) | 4 (28.5%) | | |
| C | 0 | 4 (28.5%) | | |
| D | 1 (10%) | 0 | | |
| **Child-Pugh Score** | | | Child-Pugh not available | |
| A | 7 (70%) | 6 (43%) | | |
| B | 2 (20%) | 5 (36%) | | |
| C | 1 (10%) | 3 (21%) | | |

[a]Barcelona Clinic Liver Cancer Staging.

## Facilitators of HCC care

**Physician knowledge about HCC.** Receipt of care from a physician with expertise in liver disease seemed to facilitate prompt HCC care. Some patients were under the care of a gastroenterologist who was able to quickly connect them with a multidisciplinary HCC team.

**Clear, honest, timely communication.** Most participants described interactions with their providers in a very positive way. They shared that their expectations for provider communication start with an honest, straightforward conversation. Participants shared the expectation of receiving information promptly and having appointments scheduled in a sensible time frame. In some cases, use of online patient portals served as a facilitator.

**Prognosis conversations are difficult but necessary.** Participants acknowledged that prognosis and survival conversations are not easy but still expect their providers to be clear and honest with them about these topics.

**Strong social support.** Most participants described having strong sources of support to lean on during their cancer journey. The kinds of support participants described as important included transportation, attending doctor's appointments, financial assistance, emotional support, sources of information, support for medication adherence, and assistance with domestic chores and personal care. Sources of support varied, though most included family members.

**Financial support.** Participants described a number of ways they have paid for their treatment, and most included some support from Medicaid and/or Medicare. Medicaid was often supplemented by some type of charity care through hospitals and support from family and friends. Some participants received government assistance or assistance from family and friends for non-medical bills, and this alleviated general financial worries during treatment. Some participants had private insurance that provided good coverage for medical care and sometimes supplemented Medicare coverage.

**Table 2. Key themes and sub-themes derived from patient perceptions on HCC diagnosis, facilitators and barriers to care, and suggestions for improvement.**

| Theme | Quote |
|---|---|
| **HCC Diagnosis** | |
| Receiving the Initial News–Surprise | "I went there for back pain, and they did a scan, or whatever that thing is that you go through a tunnel. They recommended that I see the liver doctor, and stuff like that. That's how it started." (UNC)<br><br>"It was a surprise and it was interesting that the brand new kidney doctor spotted the ambiguity in the ultrasound and. . .It was a surprise to me. Obviously, he wanted to see what the ambiguity was, so he ordered an MRI and the CT scan." (RCA) |
| Receiving the Initial News–Confusion and Irritation | "It pissed—it made us mad, because she had heart stents put in, what, a couple years? They were giving her the liver panel tests and everything, because some of the medicine that she was taking, they didn't want that to affect her liver. Okay. I thought she was clean as a bird. . .Yeah, the liver panel, it would show if something—maybe it's just for the medication, but I thought for sure if she had cancer, it would show that. Apparently, it didn't." (UNC)<br><br>"When Dr. ___ saw the results of those tests that were done several weeks before I saw him, his response was, which is what my response was—is that, 'Why was this not called sooner? This test here should've been an obvious red flag right here.' Of course, it was. It was off the charts, considering what they consider to be normal for cancer." (UNC) |
| **Facilitators of HCC Care** | |
| Physician Knowledge about HCC | "He arranged an appointment with Dr. B, who, as you know, is the head of the liver clinic. He was amazing. He spent almost two hours with me, talking about different possibilities of how these things can be treated. Which generated the next appointment with Dr. C and D, and their team." (UNC)<br><br>"I think the doctors down there understand more of what's going on. . . They came in and told me—he's the one who told me about the procedure, how it worked, what the goal was trying to do." (UNC) |
| Clear, Honest, Timely Communication | "Anyway, she was very straightforward, very pleasant, really—I respected her for being so honest with me. When I had had other doctors who, like I said, beat around the bush and tip-toed around the issue and never—they never wanted to commit to have the conversation. That concerned me a lot. I was very pleased." (RCA)<br><br>"Well, we just talked straight up about everything. They just told me straight up what they expected and what they required." (UNC) |
| Prognosis Conversations are Difficult but Necessary | "Probably I would like to know prognosis or how long I have. That's one thing I would wanna know because I would make sure I hit a few more things on my bucket list." (UNC)<br><br>"If I've got a question about something, if I have issues, if I have anything going on, I pick up the phone or I pick up my phone and get on MyChart and I send them a question. I can do appointments. I can do refills. I can do whatever. I always get a very quick response. It's a really, really good system." (RCA) |
| Strong Social Support | "My daughter works and she took off work to take me over there, be with me during the operation, and my wife, and then came back to take us home. My wife is very supportive. We've been married 49 years and she's just at my side." (RCA)<br><br>"I think it is a good idea to bring my brother, another set of ears and eyes. . . 'cause he thinks a little different than me. He asks something that I wouldn't think to ask."(UNC) |

*(Continued)*

**Table 2.** (Continued)

| Theme | Quote |
|---|---|
| Financial Support | "I haven't given it that much thought. I had Medicare and Medicaid. . .Oh no, no. That part wasn't no stress for me." (UNC)<br>"Well, honestly, let me tell you this. It's not impacted me near as much as it has my mother. . .She paid for everything throughout the whole deal. . .It [money] was never an issue, 'cause even if I didn't have Medicaid, that was not an issue. The issue was strictly my quality of life. My mother agreed with me 100 percent. It wasn't about the money." (RCA) |
| **Barriers to HCC Care** | |
| Provider Lack of Knowledge about HCC | "They didn't think it was cancer at the time because I had been in a car accident in 2014. The doctor then, he told me that I might've injured something in the car accident. Dr. ___, he ordered a biopsy. The biopsy turned out cancerous." (RCA) |
| Delays in Communication or Scheduling | "To see a doctor, you have to go through a bunch of hoops. . .that's very difficult when you know that you have a disease that you're fighting for your life and you have a question about it or you have an issue that comes up, and then you have to try to get this doctor to give you a referral. Then the referrals have to go through their parent company, which is the money side of things, to get approved and all this other stuff. . .it becomes very monotonous. It's very disappointing sometimes, the hoops that you have to jump through, to get results." (RCA)<br>"This ain't right. They tell you have cancer, and you can't get in contact with no one?" (UNC) |
| Poor Communication with the Medical Team | "Yes, because my primary doctor is worthless. We're in the process of changing. Terrible. When I had all of this going on, including the blood poisoning, I saw her once. She has gotten all of the results of all of the tests as they were—never once called me to talk to me or to say, 'You ought to come in to see me. We ought to talk about this.'" (RCA)<br>"I've learned that the doctors, you really got to get specific with them, because they're giving you a generalization. I need to ask more specific questions to get the specific answers that I want." (UNC) |
| Access to and Cost of Transportation | "Maybe the social worker could get more involved somehow, because transportation is an issue for me. I've called social services and asked them about some kind of transportation. They told me they didn't offer anything. . .It goes back to putting fuel in somebody's car and paying for that doggone parking. Man, it's $10.00 every time I go." (UNC) |
| Cost of Care | "Do I get to pay on my power bill, or do I go get these shots, or do I pay on what Medicare doesn't cover for these procedures that I've been having? Do I pay them. . .or do I survive?" (RCA)<br>"She's been trying to get me in with a liver doctor, but I didn't have no insurance. A specialist just don't—will not see you hardly without no insurance or without you paying for it, which I think is tough. I mean, that's just—not everybody can afford insurance these days." (UNC) |
| **Participant Suggestions for Improvement** | |
| Improve Timeliness and Coordination of Appointments | "They're so slow. It takes two weeks to do anything, anymore. . .That might be something psychological to us, because when you get in a situation like this, you want it done now. After a while, two days in your mind, you think it's actually a week, because you want it done so fast. It plays on your head a little bit." (UNC)<br>"It would be nice if they could coordinate their—have three appointments, so I don't have to come back three different days. I could do 'em all in one day. They don't try to—what do you call it—coordinate the appointments for you." (RCA) |

(*Continued*)

**Table 2.** (Continued)

| Theme | Quote |
|---|---|
| Designate a Main Point of Contact | "I guess probably the simplest thing is havin' whoever is in charge of your cancer care, this Doctor ___, havin' her nurse or whatever be the one to make sure you get everything done, scheduled, explained to you what's goin' on. . .I mean I ain't gotta talk to a doctor, but somebody that knows what's goin' on with my stuff." (UNC) |

RCA, Rapid Case Ascertainment; UNC, University of North Carolina.

## Barriers to HCC care

**Provider lack of knowledge about HCC.** Some patients initially received care from providers who, from the patient perspective, did not recognize the diagnosis immediately. Others learned of their HCC diagnosis from someone who was unable to answer their questions about treatment options and did not receive prompt referral to a subspecialist who could discuss the diagnosis and treatment options.

*Delays in Communication or Scheduling.* Participants experienced delays in communication from providers or in scheduling that, in some cases, resulted in treatment delays. Referrals from one physician to another were often a source of delay.

**Poor communication with the medical team.** One participant shared that he did not feel like providers had been honest or straightforward and wished they were more truthful about his situation and more direct in following up on what providers say they are going to do. Others felt that their providers were not proactive or specific enough in communicating results or treatment plans.

**Access to and cost of transportation.** Lack of transportation and transportation costs are a barrier for some, and overcoming that barrier can prove challenging. Resources for those with transportation issues were not readily apparent to some patients.

**Cost of care.** Some participants described medical costs as a barrier to receiving care. Some described decisions that they have faced between paying other bills and paying for treatment, while others have been unable to see a liver specialist or pay for medications due to cost.

## Participant suggestions for improvement

**Improve timeliness and coordination of appointments.** Participants would like appointments to be scheduled in a timelier and better coordinated manner to reduce their travel burden and clinic wait times.

**Designate one main point of contact.** Participants expressed a desire to have one main person to coordinate the care and be the point of contact for patients. Patients who identified one of their providers as their primary point of contact expressed greater satisfaction with their overall care.

## Discussion

This study reveals several key reasons that many patients with HCC do not receive cancer-directed treatment. On a patient level, having strong social support and access to resources such as transportation are key elements of receiving HCC care. While healthcare systems cannot create social support for those who do not have it, they can offer resources such as transportation vouchers or coordination of social services to help remove transportation as a barrier to care [26]. Reimbursement of treatment-related out-of-pocket costs (such as gasoline,

parking, and lodging) has been effective in increasing clinical trial participation [27], and similar strategies could be employed to facilitate standard-of-care HCC treatment. Additionally, caregivers of cancer patients play an essential role, providing emotional support as well as assistance with medication adherence, coordination of appointments, and communication with medical providers [28]. Leveraging these caregivers as members of the care team may be a key step in improving treatment rates in HCC, and starting with simple interventions such as assisting patients and their families with registering for online medical portals at the time of clinic intake could have considerable impact on patient-provider communication and quality of care [29].

On the healthcare system level, the perspectives of the study participants highlight areas of critically needed improvement in HCC care and our healthcare systems as a whole. While physician lack of knowledge about HCC may initially contribute to delays in diagnosis and treatment [19,23], perhaps the bigger issue is the difficulty in getting a patient connected with a knowledgeable HCC specialist. The complex process of extensive diagnostic testing and referrals to multiple providers at multiple institutions presents a challenge to patients and providers alike. These patient-described "hoops" they must jump through to access needed care are perhaps too great a hurdle for some to overcome—particularly if they have limited financial means, social support, or health literacy. The fact that the majority of participants in our study had incidentally diagnosed HCC and were not undergoing recommended routine screening suggests that some of these barriers to subspecialty care exist even upstream of the cancer diagnosis, contributing to low awareness and uptake of recommended screening ultrasounds in at-risk adults with cirrhosis [30,31]. In breast cancer, where delays in care following abnormal mammography have contributed to racial and socioeconomic disparities in patient outcomes [32], patient navigation performed by nurses, community health workers, or other members of the healthcare team has reduced barriers to care and increased timely follow up [33,34]. Patient navigation for HCC starting at the time of diagnosis might similarly improve treatment rates [35] and would provide the "clear point of contact" that patients seek and specifically mentioned in our interviews. We envision a system that leverages our cancer registrar RCA system to identify new HCC cases across the state in real time, prompting patient outreach by an HCC navigator who could facilitate specialty care, assess barriers to care, refer to health system and community-based resources, and improve general understanding of the diagnosis and requisite next steps in care. Such a navigator could be based at a specialty center but receive RCA referrals from across the state to improve access to care for all patients. This would leverage several facilitators identified in this study: the importance of access to specialists who understand HCC; the need for clear, honest, and timely communication; and the need for social and financial support. Additionally, a virtual tumor board for HCC has been shown to improve the quality and timeliness of multidisciplinary evaluation without increasing the travel burden to patients [23]. Establishing virtual tumor boards in other regions would be an evidence-based approach to improving HCC care by extending the reach of HCC specialists to communities in greatest need.

The finding of financial barriers to care for some patients was not surprising. A growing body of evidence demonstrates that financial toxicity limits access to care, treatment adherence, and patient outcomes [36–42]. What did surprise us is that the majority of the patients in our study did not cite cost of care as a barrier or burden. Many participants were enrolled in the UNC Charity Care Program, a form of financial assistance that helps ensure that poor patients receive medically necessary care. Connecting patients with available financial resources is a critical step in improving access to HCC treatment [43], and is something that could fall under the purview of a new statewide HCC navigator. Recognizing that all healthcare systems do not have the resources that can be offered in a large, public healthcare system, this

underscores the importance of easing the referral process from outside providers to tertiary care centers.

On the provider level, our participants again remind us that one of our greatest charges (and challenges) as healthcare providers is to provide clear, honest, and timely communication with our patients and their families. As our participants suggested, having a clear point of contact, such as a nurse navigator, on the clinical team who "knows what's goin' on" (direct quote from participant) and is available to answer questions, relay results, and coordinate care can facilitate the clear, honest, and timely communications that patients expect and deserve. Utilization of online portals may enhance this communication for some patients, but others will still rely on telephone communication and need to know how and whom to contact when they need something [44]. Proactive campaigns to improve 24/7 access to and utilization of nurse triage lines, such as the one implemented by Oncology Hematology Care [45], may increase access to HCC care and also reduce costly emergency department visits and hospitalizations. Increased access to specialty care through improved care coordination and navigation programs may also improve patient and caregiver understanding of diagnosis, treatment options, and prognosis [46], leveraging other key facilitators identified in this study.

A key limitation of this study is that we primarily captured the perspectives of those HCC patients who, despite the challenges they describe, were ultimately able to navigate the maze of the healthcare system to receive HCC care. We tried to mitigate this limitation in our study design by using the RCA to identify patients with a new HCC diagnosis from a variety of settings across the state, who may or may not have received any cancer care. While we attempted to include untreated patients through use of the RCA, nearly all of those who agreed to participate were receiving care for HCC. Using the RCA enabled us to include patients not cared for at an academic medical center, and their perspective is essential to understanding barriers to care. Even the patients who were able to access care faced considerable frustrations and self-reported "hoops to jump through" due to a clumsy healthcare system which often fails to meet the needs of our patients. We learned valuable lessons from their experiences that are translatable to the general HCC population and can help us better understand where some patients may be facing barriers. The fact that even those patients with self-reported financial limitations can overcome that barrier in a well-integrated health system is a key finding and should serve as an impetus to augment financial navigation and care coordination. The finding that even patients with the best of circumstances (i.e., extensive financial resources and strong social support) experienced distress and delays in care due to poor coordination of appointments and unclear communication should further serve as a charge to healthcare systems to find ways to improve.

The participants in this study are likely different from non-participants, as suggested by their varying demographic profiles, and the non-participants may face different barriers to care. While our study sample is small, interviews were conducted until thematic saturation was reached, as is the gold standard for determining sample size in interview-based qualitative research [47]. It is worth noting that one of the major challenges to recruitment for this study was difficulty reaching eligible patients by phone. Some of the barriers to participation in this study (e.g. not having a working phone, missed appointments) may also serve as barriers to receiving medical care, and strategies to mitigate these barriers should be considered in any intervention aimed at increasing HCC treatment rates.

This study highlights several important implications for HCC care and future research. First, investment in system-based supports such as caregiver engagement, patient care coordination, and financial navigation is essential. These supports are best offered in the context of a multidisciplinary subspecialty care team, as this has already been shown to improve outcomes in HCC [8,18]. Interventions to improve access to and outreach for subspecialty HCC care

might include a statewide HCC navigator, increased staffing at tertiary care centers with improved time from referral to consultation, and virtual tumor boards. Second, the impact of such interventions should be measured through outcomes such as time from diagnosis to first treatment. Finally, population-based studies of patterns of HCC treatment will be needed to assess whether these interventions improve the percentage of patients receiving any treatment for HCC over time.

## Conclusion

This qualitative evaluation of barriers and facilitators to HCC care in North Carolina should serve as the basis for tailored interventions aimed at improving access to appropriate, life-prolonging care for patients with HCC.

## Supporting information

**S1 Appendix. Semi-structured interview guide.**
(PDF)

**S2 Appendix. Codebook.**
(PDF)

## Acknowledgments

The authors wish to thank the North Carolina Central Cancer Registry Rapid Case Ascertainment team and Dr. Maihan Vu of the UNC CHAI Core for their contributions to this work and the patients and their families who participated in this study.

## Author Contributions

**Conceptualization:** Emily M. Ray, Hanna K. Sanoff.

**Data curation:** Randall W. Teal, Jessica Carda-Auten, Hanna K. Sanoff.

**Formal analysis:** Emily M. Ray, Randall W. Teal, Jessica Carda-Auten, Hanna K. Sanoff.

**Funding acquisition:** Hanna K. Sanoff.

**Investigation:** Randall W. Teal, Jessica Carda-Auten, Hanna K. Sanoff.

**Methodology:** Emily M. Ray, Randall W. Teal, Jessica Carda-Auten, Hanna K. Sanoff.

**Project administration:** Erin Coffman.

**Resources:** Emily M. Ray, Hanna K. Sanoff.

**Writing – original draft:** Emily M. Ray.

**Writing – review & editing:** Randall W. Teal, Jessica Carda-Auten, Erin Coffman, Hanna K. Sanoff.

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
