## [Decision Letter · Decision Letter 0]

20 Mar 2023

PONE-D-22-34719

Qualitative evaluation of barriers and facilitators to hepatocellular carcinoma care in North Carolina

PLOS ONE

Dear Dr. Ray,

Thank you for submitting your manuscript to PLOS ONE - it has taken us a while to find expert reviewers willing to review your work, but we have now received all comments. After careful consideration, we feel that the manuscript has merit but does not fully meet PLOS ONE’s publication criteria as it currently stands. Therefore, we invite you to submit a revised version of the manuscript that addresses the points raised during the review process.

As you will notice, the reviewers expressed a preference that the methods should be expanded upon, in order to better guide the reader - I encourage you to take up this suggestion. I also agree with reviewer 2 that further discussion on facilitators/enablers and the possible interventions they support may be particularly relevant in ensuring uptake of your work, and I would suggest to be clear in which interventions should be implemented by whom. 

We look forward to receiving your revised manuscript.

Kind regards,

Rafael Van den Bergh

Academic Editor

PLOS ONE

Journal Requirements:

Additional Editor Comments:

* Please add a description of the consent procedure into the methods section of the manuscript, and clarify why only oral consent was requested and whether it was witnessed/documented in any way.

* Please move the ethics review/approval to a separate paragraph in the methods section.

Reviewers' comments:

Reviewer's Responses to Questions

**Comments to the Author**

1. Is the manuscript technically sound, and do the data support the conclusions?

Reviewer #1: Yes

Reviewer #2: Yes

2. Has the statistical analysis been performed appropriately and rigorously? 

Reviewer #1: N/A

Reviewer #2: N/A

3. Have the authors made all data underlying the findings in their manuscript fully available?

Reviewer #1: No

Reviewer #2: No

4. Is the manuscript presented in an intelligible fashion and written in standard English?

Reviewer #1: Yes

Reviewer #2: Yes

5. Review Comments to the Author

Reviewer #1: Thank you for the opportunity to review this manuscript. It is an important topic and patients' perspectives of care experiences are crucial in developing appropriate improvements in care delivery at both system and clinician level.

- Overall: Please review the manuscript for statements that require references. See, for example, Line 88, Lines 126-7, Lines 237-238. and Lines 244-251

- Overall: Please review the manuscript for use of double quotation marks. If these are quotes from participants or from the literature on this topic, they should be identified as such. If these are not quotes from participants or from the literature, then please put them in singular inverted commas to identify them as your use of colloquial language.

- Patients and Methods [Participants and Recruitment]: Your recruitment strategy requires more explanation to help readers understand the rationale: you state you recruited until thematic saturation was reached (Lines117-118) but if you continued to recruit, you may have come across new themes in the new data.

- Patients and Methods [Data Collection Procedures]: It would be valuable to be able to see the semi-structured interview guide as an appendix or supplementary file.

- Patients and Methods [Analysis]: Please indicate the foundations of the thematic analysis approach you took. For example, your description of your method appears to bring together a quantitative (positivist - see Boyatzis 1998 and Joffe 2011) and qualitative (interpretative - see Braun & Clark 2006) approach - you might refer to it as Applied Thematic Analysis (see Guest et al. 2012 & 2014).

- Patients and Methods [Analysis]: Please provide more detail about what was in the codebook. You state "They developed topical codes from the guides" (Line 136). What were the guides?

- Patients and Methods [Analysis]: Please add a statement about how you are reporting the results. What constitutes a 'theme' in this study? Qualitative analysis is not necessarily about who said what most often, and it is in keeping with qualitative methodology to report results from the analysis of singular transcripts, but you have described thematic saturation, so it would be helpful to readers if you explained your choices. This is particularly important given that in the Results section you refer to "most", "some" and "other" in relation to participants' perceptions and opinions.

- Results [Participant Suggestions for Improvement/ Improve Timeliness and Coordination of Appointments]: Please explain or delete the interpretive statement: "seemed to have greater satisfaction with their overall care." On what basis have you concluded "seemed to"?

- Discussion: This is a well developed discussion of the themes. It leans towards advocacy in tone, so please review this section to ensure that all statements are derived from the findings or are referenced.

Reviewer #2: General comments:

This is a well written article with a clearly described aim and methods. I have only minor comments and suggestions.

Specific comments:

1. Page 10, Line 112 – the authors mention they purposively sampled participants – what was the sampling frame (e.g., characteristics) that was used?

2. Please describe the conceptual framework used to guide the study, and specifically the interview guide and analysis.

3. I feel the that the results would benefit from a content analysis approach in the reporting – for example, it would be useful to know how many ‘codes’ arose for each of the subthemes e.g., 7 participants mentioned something about ‘improved timeliness and coordination of appointments’. Percentages are not necessary as they would not be useful for qualitative analysis, but the numbers of patients who reported would give good context.

4. Some aspects of the results need more insight and explanation (e.g., Page 10, Lines 179-181; Page 12, Lines 220-222) – one line to describe a subtheme seems a little thin – is there any further insight that can be provided?

5. Page 14, Line 252 – the authors mention routine screening for HCC and that majority of patients were not undergoing routine screening. Can they elaborate on this more? Is routine screening for HCC available in the US (e.g., similarly to breast cancer or colorectal cancer screening)?

6. Discussion – the authors summarise and critique the barriers to care well, but I would ask for some more insight into the facilitators and how these might be used as a basis for interventions (see next point).

7. In the Abstract, the purpose of the study is to inform “tailor[ing] interventions to increase access to appropriate therapy” – could the authors please provide some more insight in the Discussion as to how their study findings will be used to tailor or develop such interventions?

6. PLOS authors have the option to publish the peer review history of their article (what does this mean?). If published, this will include your full peer review and any attached files.

Reviewer #1: **Yes: **Klay Lamprell

Reviewer #2: **Yes: **Rebecca Venchiarutti

---

## [Editor Report · Decision Letter 1]

5 Jun 2023

Qualitative evaluation of barriers and facilitators to hepatocellular carcinoma care in North Carolina

PONE-D-22-34719R1

Dear Dr. Ray,

We’re pleased to inform you that your manuscript has been judged scientifically suitable for publication and will be formally accepted for publication once it meets all outstanding technical requirements.

Kind regards,

Rafael Van den Bergh

Academic Editor

PLOS ONE
---

## [Editor Report · Acceptance letter]

13 Jun 2023

PONE-D-22-34719R1 

Qualitative evaluation of barriers and facilitators to hepatocellular carcinoma care in North Carolina 

Dear Dr. Ray:

I'm pleased to inform you that your manuscript has been deemed suitable for publication in PLOS ONE. Congratulations! Your manuscript is now with our production department. 

Kind regards, 

on behalf of

Dr. Rafael Van den Bergh 

Academic Editor

PLOS ONE